# Graph GOSPA similarity function for Gaussian Process regression on graphs

## Abstract

In this paper we propose a similarity function between graphs based on a mathematically principled metric for graphs of different sizes: the graph generalised optimal subpattern assignment (GOSPA) metric. The similarity function is based on an optimal assignment between nodes and has an interpretable meaning in terms of similarity for node attribute error, number of unassigned nodes, and number of edge mismatches. The proposed similarity function is computable in polynomial time. We also propose its use in Gaussian processes (GPs) for graphs to predict molecular properties. Experimental results show the benefits of the proposed GP model compared to other GP baselines.

## 1 Introduction

With the success of machine learning in multiple research areas, data-driven analysis plays a more important role in many applications in chemistry, including prediction of chemical properties Delaney (2004); Lusci et al. (2013); Mobley et al. (2014); An et al. (2024), chemical reactivities Coley et al. (2019) and drug discovery von Lilienfeld & Burke (2020); Ahn et al. (2021). Recent advances in deep neural networks (DNNs) have demonstrated promising performance in many tasks and have been widely used in molecular property prediction tasks Yang et al. (2019); Chithrananda et al. (2020); Chen et al. (2018); Meuwly (2021). However, to make a DNN model successful, high-quality and comprehensive datasets are the key. This drawback becomes important when exploring a new class of molecules, as a limited quantity of high-quality experimental data is usually available in the very early stage of the exploration Thawani et al. (2020).

Gaussian processes (GPs) are a type of kernel-based method to solve regression and classification problems Rasmussen & Williams (2006) that are specially suitable for small datasets, since they typically only have few parameters. GPs can be used with inputs that are graphs using a kernel for graphs Nikolentzos et al. (2021). Generally, there are three types of kernels or similarity functions for graphs:

(I) Diffusion kernels based on a metric on graphs (Neuhaus & Bunke, 2007, Chap. 5), such as the graph edit distance (GED) Sanfeliu & Fu (1983). A drawback of these kernels is that they are computationally intensive to compute because of the matrix exponential in kernel calculation.

(II) Similarity measures based on applying a transformation to the GED such that low metric values are mapped to high similarities, and the other way round (Neuhaus & Bunke, 2007, Chap. 5). While these transformations do not define valid kernels, they can be used in practice Boughorbel et al. (2004). A drawback is that the computation of the GED is generally NP-hard Zeng et al. (2009).

(III) Kernels based on features obtained via pre-processing of the graphs, which can imply a loss of information. Examples of these are the random walk kernel Kashima et al. (2003); Gardner et al. (2018), and the Weisfeiler-Lehman (WL) graph kernel Shervashidze et al. (2011); Griffiths et al. (2024).

In this paper, we propose a similarity function between graphs, where each node can have certain features, that is based on a mathematically principled metric for graphs, meeting the identity, symmetry and triangle inequality properties. In particular, we propose to use the graph generalized optimal subpattern assignment (GOSPA) metric Gu et al. (2024); Rahmathullah et al. (2017). The graph GOSPA metric is based on computing an optimal assignment between nodes by penalising node attributes

for assigned nodes, the number of unassigned nodes and the number of edge mismatches. Therefore, the graph GOSPA similarity function has an interpretable meaning, inherited from the graph GOSPA metric, which takes into account the whole information of the graph and can be computed in polynomial time.

Our contributions can be summarised as follows:

(1) We propose a novel similarity measure for graphs, based on the graph GOSPA metric.

(2) We show the decomposition of the graph GOSPA similarity into interpretable components.

(3) We use the graph GOSPA similarity as the kernel function of a GP to predict molecular properties in several datasets. Experimental results demonstrate that Graph GOSPA GP has the best performance compared to other GP baselines in several of the considered datasets. We also show that the decomposition of the kernel can be used to assist with the interpretation of the similarity score.

## 2 Background on graphs and graph GOSPA metric

### 2.1 Weighted undirected graphs

A weighted, undirected graph is formed by vertices (also called nodes) and weighted edges, each edge connecting two vertices. The set of vertices is $V = \{x_1, \ldots, x_n\}$ with the $i$-th node feature denoted by $x_i \in \mathbb{R}^N$ Trudeau (1993). The edges and their weights can be represented by a symmetric adjacency matrix $A \in \mathbb{R}^{n \times n}$, whose $(i, j)$ element $A(i, j)$ indicates the weight between the $i$-th and $j$-th node, with $A(i, j) = 0$ indicating no edge.

### 2.2 Graph GOSPA metric

The graph GOSPA metric is a mathematically principled metric, as it meets the identity, symmetry, and triangle inequality properties, for graphs of different sizes Gu et al. (2024). Let us consider two graphs $X$ and $Y$ with vertices $V_X = \{x_1, \ldots, x_{n_X}\}$, $V_Y = \{y_1, \ldots, y_{n_Y}\}$, and adjacency matrices $A_X \in \mathbb{R}^{n_X \times n_X}$ and $A_Y \in \mathbb{R}^{n_Y \times n_Y}$.

The graph GOSPA metric looks for an optimal assignment between nodes in $V_X$ and nodes in $V_Y$, but it can leave some nodes unassigned. The assignments between $V_X$ and $V_Y$ can be represented by a binary matrix $(n_X + 1) \times (n_Y + 1)$. We use $\mathcal{W}_{X,Y}$ to denote the set of all binary matrices. A matrix $W \in \mathcal{W}_{X,Y}$ satisfies:

$$\sum_{i=1}^{n_X+1} W(i, j) = 1, \; j = 1, \ldots, n_Y \tag{1}$$

$$\sum_{j=1}^{n_Y+1} W(i, j) = 1, \; i = 1, \ldots, n_X \tag{2}$$

$$W(n_X + 1, n_Y + 1) = 0, \tag{3}$$

$$W(i, j) \in \{0, 1\}, \, \forall \, i, j \tag{4}$$

The element $W(i, j) = 1$ if $x_i$ is assigned to $y_j$. If $x_i$ remains unassigned, $W(i, n_Y + 1) = 1$, and if $y_j$ remains unassigned then $W(n_X + 1, j) = 1$.

If we consider $X$ to be a ground truth graph and $Y$ an estimate (obtained by some algorithm), the unassigned nodes in $X$ and $Y$ are referred to as missed and false nodes, respectively.

**Definition 1.** *For $1 < p < \infty$, a scalar $c > 0$, edge mismatch penalty $\epsilon > 0$ and base metric $d(\cdot, \cdot)$ on the node feature space $\mathbb{R}^N$, the graph GOSPA metric $d_p^{(c,\epsilon)}(\cdot, \cdot)$ between two graphs $X$ and $Y$ is*

$$d_p^{(c,\epsilon)}(X, Y) = \min_{W \in \mathcal{W}_{X,Y}} \left( \mathrm{tr}\big[D_{X,Y}^\top W\big] + e_{X,Y}(W)^p \right)^{1/p} \tag{5}$$

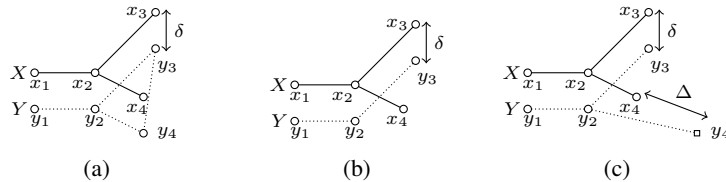

(a)                          (b)                          (c)

Figure 1: Example to illustrate the node and edge mismatch costs for the same ground truth graph $X$, and different estimated graphs $Y$. (a) All nodes are properly assigned and one edge mismatch; (b) three properly assigned nodes, one missing node and a half edge mismatch penalty; (c) three properly assigned nodes, two unassigned nodes and two half-edge mismatch penalties.

*where*

$$D_{X,Y}(i,j) = \begin{cases} d\left(x_i, y_j\right)^p & i \leq n_X, j \leq n_Y, \\ \frac{c^p}{2} & i = n_X + 1, j \leq n_Y, \\ \frac{c^p}{2} & i \leq n_X, j = n_Y + 1, \\ 0 & i = n_X + 1, j = n_Y + 1, \end{cases} \tag{6}$$

*and*

$$e_{X,Y}(W)^p = \frac{\epsilon^p}{2}||A_X W_{1:n_X, 1:n_Y} - W_{1:n_X, 1:n_Y} A_Y||, \tag{7}$$

*where $W_{1:n_X, 1:n_Y}$ is the matrix formed by the first $n_X$ rows and the first $n_Y$ columns of matrix $W$ (e.g., removing the last row and column of $W$) and $|| \cdot ||$ is the component-wise 1-norm of a matrix.*

Due to the binary constraint in (4), it is NP hard to compute (5). With the relaxation of the constraints in (4) to $W(i,j) \geq 0, \forall i, j$, we obtain a relaxed version of the metric, which also satisfies the metric properties and can be computed in polynomial time using linear programming Khachiyan (1980). We also refer to this relaxed version of the metric as the graph GOSPA metric.

The graph GOSPA metric penalises node attribute errors for assigned nodes, number of unassigned nodes (each with a cost $c^p/2$), and number of edge mismatches. In particular, for two pairs of assigned nodes (two nodes in $X$ and two nodes in $Y$), the edge mismatch penalty is $\epsilon^p$ multiplied by the absolute difference in the corresponding edge weights. In addition, each edge connecting an assigned node and an unassigned node creates a half-edge mismatch penalty of $\epsilon^p/2$ multiplied by the weight of the edge, see full details in Gu et al. (2024).

### 2.3 EXAMPLES

We illustrate how the graph GOSPA metric works using the examples in Figure 1. This figure compares a ground truth graph $X$ with three different graph estimates $Y$. The weights of all edges are one. In all these examples, we consider that the distance between all assigned nodes is $\delta$, $p = 1$, $\epsilon \ll c$, and $\delta \ll c$.

In Figure 1a, we have the ground truth graph $X$ with nodes $x_1$, $x_2$, $x_3$ and $x_4$, denoted by circles, and graph $Y$ with nodes $y_1$, $y_2$, $y_3$ and $y_4$. In graph $Y$, we also use circles to denote the nodes assigned to the corresponding nodes in graph $X$. Although the nodes in $Y$ are all assigned to the nodes in $X$, there is an extra edge between node $y_3$ and $y_4$ which does not exist in graph $X$. Thus, the distance between graph $X$ and graph $Y$ only has note attribute (localisation) errors $\delta$ for each assigned node, and one edge mismatch error $\epsilon^p$. The metric value is $d_p^{(c,\epsilon)}(X,Y) = 4\delta + \epsilon$.

In Figure 1b, we compare graphs of different sizes. There is one node missing in graph $Y$, which leaves node $x_4$ unassigned. In this case, apart from the localisation errors in the assigned nodes, there is also a penalty for the unassigned node. Furthermore, there is an edge connected to $x_4$ in graph $X$, and this contributes with a half-mismatch penalty. The metric value is $d_p^{(c,\epsilon)}(X,Y) = 3\delta + \frac{c}{2} + \frac{\epsilon}{2}$.

In Figure 1c, both graphs have four nodes, and we consider that $\Delta \gg c$. This implies that node $x_4$ and node $y_4$ are unassigned, and they contribute to two unassigned node errors. There is one edge connected to each unassigned node, that is also connected to an assigned node on the other end, so they both contribute to a half-edge mismatch error. Thus, the distance consists of localisation errors, two unassigned node errors, and two half-edge mismatch errors. The metric value is $3\delta + c + \epsilon$.

## 3 GAUSSIAN PROCESSES FOR GRAPHS WITH GRAPH GOSPA SIMILARITY

### 3.1 GAUSSIAN PROCESSES

A Gaussian Process (GP) is a non-parametric Bayesian model over functions Rasmussen & Williams (2006). A GP can be fully specified by its mean $m(\cdot)$ and covariance function (also called kernel) $k(\cdot, \cdot)$ and we can write it as $f \sim \mathcal{GP}(m(\cdot), k(\cdot, \cdot))$. The mean function $m(\cdot)$ is typically chosen to be a constant function with zero value, while the choice of covariance function given by that kernel may vary from task to task.

For a regression task, consider that we have a set of $n$ data points, $\mathcal{D} = \{(\mathbf{x}_i, y_i)\}_{i=1}^n$, where $\mathbf{x}_i \in \mathbb{X}$ is the input data point and $y_i \in \mathbb{R}$ is its associated output. Given the dataset and a new test point $\mathbf{x}_*$, we want to infer the value of its output, a problem called regression. In GP regression, we assume that there is additive noise such that $y_i = f(\mathbf{x}_i) + \epsilon_i$, where $f(\mathbf{x}_i)$ is the function value of sample $\mathbf{x}_i$ and $\epsilon_i$ is a zero-mean Gaussian noise with variance $\sigma^2$, which is independent of other variables.

For simplicity, we write the function value $f(\mathbf{x}) = [f(\mathbf{x}_1), \ldots, f(\mathbf{x}_n)]^T$ as $\mathbf{f}$, and the function value $f(\mathbf{x}_*)$ as $\mathbf{f}_*$ and $\mathbf{y} = \{y_1, \ldots, y_n\}$. The marginal likelihood of the model can be written as $p(\mathbf{y}|\mathbf{x}) = \int p(\mathbf{y}|\mathbf{f}, \mathbf{x}) p(\mathbf{f}|\mathbf{x}) \, d\mathbf{f}$, where we marginalise the function value $\mathbf{f}$. In a Gaussian processes model, the prior is assumed to be a zero-mean Gaussian, $\mathbf{f}|\mathbf{x} \sim \mathcal{N}(0, \mathbf{K})$, where $\mathbf{K}$ refers to the $n$ by $n$ covariance matrix whose $(i, j)$ element is $k(\mathbf{x}_i, \mathbf{x}_j)$. The likelihood is also a Gaussian, $\mathbf{y}|\mathbf{f} \sim \mathcal{N}(\mathbf{f}, \sigma_n^2 I)$, so the log marginal likelihood is

$$\log p(\mathbf{y}|\mathbf{x}) = -\frac{1}{2}\mathbf{y}^T(\mathbf{K} + \sigma_n^2\mathbf{I})^{-1}\mathbf{y} - \frac{1}{2}\log|\mathbf{K} + \sigma_n^2\mathbf{I}| - \frac{n}{2}\log 2\pi.$$

For the test input $\mathbf{x}_*$, the joint distribution of the observed target values and the function values at the test locations can be written as

$$\begin{bmatrix} \mathbf{y} \\ \mathbf{f}_* \end{bmatrix} \sim \mathcal{N}\left(\mathbf{0}, \begin{bmatrix} \mathbf{K} + \sigma_n^2\mathbf{I} & \mathbf{K}_* \\ \mathbf{K}_*{}^T & K_{**} \end{bmatrix}\right), \tag{8}$$

where $\mathbf{K}_* = [k(\mathbf{x}_1, \mathbf{x}_*), \ldots, k(\mathbf{x}_n, \mathbf{x}_*)] = [k(\mathbf{x}_*, \mathbf{x}_1)^T, \ldots, k(\mathbf{x}_*, \mathbf{x}_n)^T]$ and $K_{**} = k(\mathbf{x}_*, \mathbf{x}_*)$. Then the posterior of the test output, which solves the regression problem and provides its associated uncertainty is

$$\mathbf{f}_*|\mathbf{x}_*, \mathbf{x}, \mathbf{f} \sim \mathcal{N}(\mathbf{K}_*^T(\mathbf{K} + \sigma_n^2\mathbf{I})^{-1}\mathbf{y}, K_{**} - \mathbf{K}_*^T(\mathbf{K} + \sigma_n^2\mathbf{I})^{-1}\mathbf{K}_*) \tag{9}$$

### 3.2 KERNEL BASED ON THE GRAPH GOSPA METRIC

Once we have a training set, as explained above, we can apply a GP if we have a kernel function. A kernel is a function $k : \mathbb{X} \times \mathbb{X} \to \mathbb{R}$ that measures the similarity between elements of the space $\mathbb{X}$. In addition, a kernel can be written as the inner product on a feature space $\mathbb{F}$ that corresponds to the mapping $\phi : \mathbb{X} \to \mathbb{F}$. That is, given $x, y \in \mathbb{X}$, $k(x, y) = \langle \phi(x), \phi(y) \rangle$. Such projection function $\phi$ exists if and only if $k$ is a positive-semidefinite function, which means the Gram matrix $K_{i,j} = k(x_i, x_j)$, where $x_1, \ldots, x_n \in \mathbb{X}$ and $i, j \in \{0, \ldots, n\}$, is positive-semidefinite for every possible set of data points.

Kernel functions like the radial basis function (RBF) kernel and the Matérn kernels are commonly used Rasmussen & Williams (2006), but they are designed for a vector input, $\mathbf{x} \in \mathbb{R}^{\mathbb{N}}$. For graphs, there are kernels such as random walk kernels Vishwanathan et al. (2010) or Weisfeiler-Lehman graph kernels Shervashidze et al. (2011). An alternative is to define similarity functions that can work as kernels, but do not meet the above properties, for instance, a similarity function based on the GED (Neuhaus & Bunke, 2007, Chap. 5).

Here we introduce the similarity function based on the graph GOSPA metric.

**Definition 2.** *Let $X$ and $Y$ be two graphs, $p' > 1$ and $\ell > 0$, a length scale hyperparameter. We define the similarity function between two graphs based on the graph GOSPA metric $d_p^{(c,\epsilon)}(\cdot, \cdot)$ as*

$$k(X, Y) = \exp\left(-\frac{d_p^{(c,\epsilon)}(X, Y)^{p'}}{\ell}\right). \tag{10}$$

As will be shown in Section 3.3, this similarity function can be decomposed into its different components to provide clear interpretability of the results. Although trivial similarity functions defined like this are not generally positive semidefinite Vert (2008), they can show suitable performance in practice Boughorbel et al. (2004); Neuhaus & Bunke (2007). To improve the stability of the algorithm, we add $\sigma^2 \mathbf{I}$ to the covariance matrix $\mathbf{K}$. In addition, during the training process, if the resulting covariance matrix for a given choice of hyperparameters $(c, \epsilon, p, p')$ is not positive definite, these hyperparameters are discarded.

### 3.3 Decomposition of graph GOSPA similarity function

In this section, we present the decomposition of the graph GOSPA similarity function. We first review the graph GOSPA metric decomposition into different types of costs Gu et al. (2024). We know from the graph GOSPA metric that $D_{X,Y}(i,j)$ represents the following costs:

1. Node attribute (localisation) error for assigned nodes, if $i \leq n_X$, $j \leq n_Y$.
2. Missed node cost if $i \leq n_X$, $j = n_Y + 1$.
3. False node cost if $i = n_X + 1$, $j \leq n_Y$.

The sets of indices $(i, j)$ that belong to each of the previously mentioned categories are denoted by $\mathcal{S}_1$, $\mathcal{S}_2$ and $\mathcal{S}_3$. Therefore, for a given assignment matrix $W$, we have the following costs: node attribute (localisation) cost, number of missed nodes cost, and number of false nodes cost. Mathematically, these are given by

$$l(X, Y, W)^p = \sum_{(i,j) \in \mathcal{S}_1} D_{X,Y}(i,j) W(i,j) \tag{11}$$

$$m(X, Y, W)^p = \frac{c^p}{2} \sum_{(i,j) \in \mathcal{S}_2} W(i,j) \tag{12}$$

$$f(X, Y, W)^p = \frac{c^p}{2} \sum_{(i,j) \in \mathcal{S}_3} W(i,j). \tag{13}$$

Let $W^*$ denote the optimal assignment in (5). Then, the graph GOSPA metric can be written as

$$d_p^{(c,\epsilon)}(X, Y) = \left( l(X, Y, W^*)^p + m(X, Y, W^*)^p + f(X, Y, W^*)^p + e_{X,Y}(W^*)^p \right)^{1/p}. \tag{14}$$

Therefore, the graph GOSPA similarity function for $p' = p$ can be written as the product over the similarity functions for node attribute errors, number of missed nodes, number of false nodes and edge mismatches

$$k(X, Y) = k_l(X, Y)\, k_m(X, Y)\, k_f(X, Y)\, k_e(X, Y) \tag{15}$$

where

$$k_l(X, Y) = \exp\left( -\frac{l(X, Y, W^*)^p}{\ell} \right) \tag{16}$$

$$k_m(X, Y) = \exp\left( -\frac{m(X, Y, W^*)^p}{\ell} \right) \tag{17}$$

$$k_f(X, Y) = \exp\left( -\frac{-f(X, Y, W^*)^p}{\ell} \right) \tag{18}$$

$$k_e(X, Y) = \exp\left( -\frac{e_{X,Y}(W^*)^p}{\ell} \right). \tag{19}$$

It is also possible to merge the similarities for the missed and false nodes into a single similarity score for unassigned nodes, given by the product of these two similarities $k_u(X, Y) = k_m(X, Y) k_f(X, Y)$. Here we show the similarity decomposition for the examples in Figure 1.

In Figure 1a, the similarity for localisation is $k_l(X, Y) = \exp(-4\delta/\ell)$. The similarity for unassigned nodes is $k_u(X, Y) = 1$, and the edge similarity is $k_e(X, Y) = \exp(-\epsilon/\ell)$. We can see that the similarity for unassigned nodes is one, meaning that there are not any unassigned nodes in the optimal assignment. In Figure 1b, the similarity for localisation is $k_l(X, Y) = \exp(-3\delta/\ell)$, the similarity for unassgined nodes is $k_u(X, Y) = \exp(-c/2\ell)$, and the edge similarity is $k_e(X, Y) = \exp(-\epsilon/2\ell)$. In this case, none of the similarity decompositions are one, meaning that the graphs differ in the localisation of some nodes, there are unassigned nodes, and also edge mismatches. In Figure 1c, the similarity for localisation is also $k_l(X, Y) = \exp(-3\delta/\ell)$, the similarity for unassigned node is $k_u(X, Y) = k_m(X, Y)k_f(X, Y) = \exp(-c/\ell)$ and the edge similarity is $k_e(X, Y) = \exp(-\epsilon/\ell)$.

In Section 4.3, we also provide an example of how this similarity decomposition can be applied to molecules. With the decomposition of the graph GOSPA similarity, we can have a better interpretation on the similarities or dissimilarities between graphs, which can assist in the understanding of the GP predictions.

## 4    EXPERIMENTAL RESULTS

In this section, we first compare the Gaussian process based on the graph GOSPA similarity function with other Gaussian process models to make predictions on molecular properties in real datasets. Then, we illustrate the decomposition of the graph GOSPA similarity function applied to molecules.

### 4.1    EXPERIMENTAL SETUP

#### DATASETS

In the experiments, we use 6 regression datasets, five from MoleculeNet Wu et al. (2018), and one from Griffiths et al. (2022). Specifically, ESOL, FreeSolv, Lipophilicity and Photoswitch are datasets about the physical chemical properties of molecules and there is only one property to predict. QM8 is a dataset consisting of quantum mechanical properties. In this dataset, for numerical tractability of GPs, we only use a subset of the molecules by random sampling 2000 molecules from the full dataset and only considering the first 6 properties to predict.

The datasets are split into training and test sets with a ratio of 80/20 (note that validation sets are not required for GP models since hyperparameters are chosen based on the marginal likelihood objective on the training set). The graphs are obtained by converting the SMILES strings Weininger (1988) into the corresponding molecular graphs.

#### BASELINES

We compare the proposed method with GPs with the following kernels for molecules: Tanimoto kernel Ralaivola et al. (2005) using ECFP fingerprints Rogers & Hahn (2010), subsequence string kernel (SSK) Moss et al. (2020) using SMILES Weininger (1988) and WL kernel Shervashidze et al. (2011) using graphs with atom type as the node attributes. Shortest path kernel Borgwardt & Kriegel (2005) for labelled graphs, neighbourhood hash kernel Hido & Kashima (2009), edge histogram kernel and vertex histogram kernel Sugiyama & Borgwardt (2015).

#### EVALUATION METRICS

For ESOL, FreeSolv, Lipophilicity and Photoswitch datasets, we use the root mean square error (RMSE) to evaluate the performance. For QM8, we use mean absolute error (MAE), as this is the common choice in other papers for this dataset Wu et al. (2018); Hu et al. (2020). We also use negative log predictive density (NLPD) as the metric to quantify the uncertainty Griffiths et al. (2024).

#### IMPLEMENTATION DETAILS

All GP models are single-output GPs and the results are obtained by averaging over 20 random splits of the training and the test set. The node attribute in the GP model based on graph GOSPA similarity is the atom type. The base metric for node attributes of the graph GOSPA metric is $d(x, y) = 0$ if $x = y$, and $d(x, y) = c$ if $x \neq y$. All GPs are trained using the L-BFGS-B optimiser Liu & Nocedal

(1989), except the graph GOSPA similarity, which is trained with the Adam optimiser Kingma & Ba (2014) on the marginal log-likelihood with 2000 iterations. The learning rate is set to 0.001. The hyperparameters for graph GOSPA metric in the graph GOSPA similarity function are set to $c = 3$, $p = 2$, $p' = 1$ and the value of $\epsilon$ is set based on the optimal value of marginal likelihood with grid search between [0,3] with step 0.2. The hyperparameter $\ell$ for the graph GOSPA similarity function is optimised during the GP training process, the initial value is set to $\ell = 1$.

The models using SSK kernel, Tanimoto kernel and graph GOSPA similarity are implemented in GPflow [1] Matthews et al. (2017). The WL kernel, shortest path kernel, neighbourhood has kernel, edge histogram kernel and vertex histogram kernel are obtained from functions in the GraKeL library Siglidis et al. (2020). The GP models for these graph kernels are using the implementation in the library GAUCHE Griffiths et al. (2024), which are implemented in GPytorch Gardner et al. (2018).

## 4.2 RESULTS

Table 1 shows the results of the proposed methods and the baselines on the molecular datasets. The best results for each task are shown in bold, and the underlined values are the second-best results. From Table 1, it can be observed that the proposed graph GOSPA similarity performs the best in three datasets, and second best in the FreeSolv dataset.

Table 1: Molecular property prediction over 4 physical chemical datasets.

| | Dataset (RMSE ↓) | | | |
|---|---|---|---|---|
| Kernels | ESOL | FreeSolv | Lipophilicity | Photoswitch |
| SSK | 0.66 ± 0.02 | **1.34 ± 0.03** | 0.73 ± 0.01 | 26.62 ± 1.07 |
| Tanimoto | 1.02 ± 0.02 | 1.88 ± 0.13 | 0.76 ± 0.01 | 23.42± 0.80 |
| WL Kernel | 0.75 ± 0.01 | 1.48 ± 0.04 | 0.74 ± 0.01 | 24.02 ± 0.65 |
| Shortest Path Labelled | 0.98 ± 0.01 | 2.41 ± 0.05 | 1.02 ± 0.02 | 43.58 ±7.11 |
| Neighbourhood Hash | 0.96 ± 0.05 | 1.82 ± 0.13 | 1.71 ± 0.18 | 33.62 ± 5.11 |
| Edge Histogram | 2.12 ± 0.02 | 3.94 ± 0.09 | 1.19 ± 0.01 | 66.76 ± 1.10 |
| Vertex Histogram | 1.12 ± 0.01 | 2.93 ± 0.07 | 1.09 ± 0.01 | 48.95 ± 1.52 |
| Graph GOSPA | **0.66 ± 0.01** | 1.37 ± 0.05 | **0.70 ± 0.03** | **21.44 ± 0.68** |

In Table 2, which contains the results of the QM8 dataset, the SSK kernel produces the best results followed by graph GOSPA and Tanimoto. Graph GOSPA performs the best among the algorithms that use a molecular graph as input.

Table 2: Molecular property prediction over a subset of 2000 molecules on the QM8 dataset. MAE values are scaled up by 100.

| | Dataset (MAE ↓) | | | | | |
|---|---|---|---|---|---|---|
| Kernels | QM8 subset (results scaled up by $10^2$) | | | | | |
| | E1-CC2 | E2-CC2 | f1-CC2 | f2-CC2 | E1-PBE0 | E2-PBE0 |
| SSK | **1.41 ± 0.01** | **1.20 ± 0.02** | 2.46 ± 0.06 | 4.09 ± 0.04 | **1.29 ± 0.01** | 2.34 ± 0.05 |
| Tanimoto | 1.41 ± 0.01 | 1.36 ± 0.02 | **2.45 ± 0.05** | 3.98 ± 0.06 | 1.47 ± 0.01 | **2.29 ± 0.05** |
| WL kernel | 2.76 ± 0.02 | 1.99 ± 0.01 | 2.83 ± 0.03 | 4.23 ± 0.03 | 3.10 ± 0.02 | 2.51 ± 0.02 |
| Shortest Path Labelled | 2.94 ± 0.02 | 2.13 ± 0.01 | 2.93 ± 0.02 | 4.45 ± 0.26 | 3.27 ± 0.02 | 2.67 ± 0.02 |
| Neighbourhood Hash | 2.97 ± 0.03 | 2.26 ± 0.04 | 3.62 ± 0.26 | 4.57 ± 0.08 | 3.29 ± 0.05 | 2.81 ± 0.09 |
| Edge Histogram | 3.61 ± 0.03 | 2.66 ± 0.02 | 3.17 ± 0.03 | 4.66 ± 0.03 | 3.87 ± 0.03 | 3.18 ± 0.02 |
| Vertex Histogram | 3.24 ± 0.02 | 2.29 ± 0.02 | 2.97 ± 0.03 | 4.55 ± 0.03 | 3.55 ± 0.02 | 2.87 ± 0.02 |
| Graph GOSPA | 1.48 ± 0.01 | 1.29 ± 0.01 | 2.54 ± 0.06 | **3.81 ± 0.05** | 1.41 ± 0.02 | 2.44 ± 0.05 |

The NLPD results are shown in Table 3 and Table 4. In Table 3, we can observe that the GP with graph GOSPA performs the best in quantifying uncertainty in the ESOL and the Photoswitch dataset and second best in the FreeSolv and the Lipophilicity dataset. It can be observed that, in Table 4, the GP with Graph GOSPA similarity function generally outperforms the considered baselines in uncertainty quantification.

---

[1]Code will be released via Github if the paper is accepted.

Table 3: Uncertainty quantification over 4 physical chemical datasets.

| | Dataset (NLPD ↓) | | | |
|---|---|---|---|---|
| Kernels | ESOL | FreeSolv | Lipophilicity | Photoswitch |
| SSK | 0.40 ± 0.05 | **0.08 ± 0.06** | 1.18 ± 0.02 | 0.64 ± 0.07 |
| Tanimoto | 1.00 ± 0.09 | 0.62 ± 0.04 | 0.89 ± 0.02 | 0.39 ± 0.04 |
| WL Kernel | 0.38 ± 0.02 | 0.38 ± 0.02 | **0.75 ± 0.01** | 0.36 ± 0.03 |
| Shortest Path Labelled | 0.62 ± 0.01 | 0.92 ± 0.02 | 1.22 ± 0.01 | 0.39 ± 0.03 |
| Neighbourhood Hash | 1.88 ± 0.18 | 1.49 ± 0.13 | 3.57 ± 0.12 | 1.53 ± 0.16 |
| Edge Histogram | 1.41 ± 0.01 | 1.44 ± 0.03 | 1.41 ± 0.01 | 1.38 ± 0.02 |
| Vertex Histogram | 0.80 ± 0.01 | 1.16 ±0.03 | 1.33 ± 0.01 | 1.13 ± 0.04 |
| Graph GOSPA | **0.30 ± 0.08** | 0.27 ± 0.03 | 0.85 ± 0.11 | **0.33 ± 0.04** |

Table 4: Uncertainty quantification over a subset of 2000 molecules on the QM8 dataset.

| | Dataset (NLPD ↓) | | | | | |
|---|---|---|---|---|---|---|
| Kernels | QM8 subset | | | | | |
| | E1-CC2 | E2-CC2 | f1-CC2 | f2-CC2 | E1-PBE0 | E2-PBE0 |
| SSK | 3.44 ± 0.17 | 4.10 ± 0.13 | 4.46 ± 1.12 | 8.24 ± 0.40 | 3.08 ± 0.16 | 3.68 ± 0.10 |
| Tanimoto | **0.59 ± 0.01** | 0.78 ± 0.01 | 1.13 ± 0.04 | 1.24 ± 0.02 | **0.56 ± 0.01** | 0.71 ± 0.01 |
| WL kernel | 1.22 ± 0.01 | 1.34 ± 0.01 | 1.35 ± 0.03 | 1.34 ± 0.01 | 1.24 ± 0.01 | 1.19 ± 0.01 |
| Shortest Path Labelled | 1.25 ± 0.01 | 1.36 ± 0.01 | 1.38 ± 0.03 | 1.37 ± 0.01 | 1.27 ± 0.01 | 1.24 ± 0.01 |
| Neighbourhood Hash | 1.64 ± 0.12 | 1.50 ± 0.06 | 1.97 ± 0.16 | 2.21 ± 0.53 | 1.54 ± 0.12 | 1.84 ± 0.18 |
| Edge Histogram | 1.43 ± 0.01 | 1.44 ± 0.01 | 1.43 ± 0.03 | 1.40 ± 0.01 | 1.42 ± 0.01 | 1.42 ± 0.01 |
| Vertex Histogram | 1.33 ± 0.01 | 1.38 ± 0.01 | 1.41 ±0.03 | 1.39 ±0.01 | 1.34 ± 0.01 | 1.29 ± 0.01 |
| Graph GOSPA | 0.63 ± 0.01 | **0.72 ± 0.01** | **1.12 ± 0.04** | **1.20 ± 0.02** | 0.62 ± 0.02 | **0.66 ± 0.01** |

### 4.3 Decomposition of graph GOSPA similarity example

In this section, we illustrate how the graph GOSPA similarity function can be decomposed into different parts to quantify the similarity of different parts in a graph (node attributes, unassigned nodes and edge mismatches). For demonstration, we choose three molecules from the ESOL dataset, shown in Figure 2. We set the hyperparameters $c = 3$, $p' = p = 1$, $\epsilon = 0.8$, and $\ell$ has been set to the optimised value on the ESOL dataset, $\ell = 27.371$, see Section 4.1.

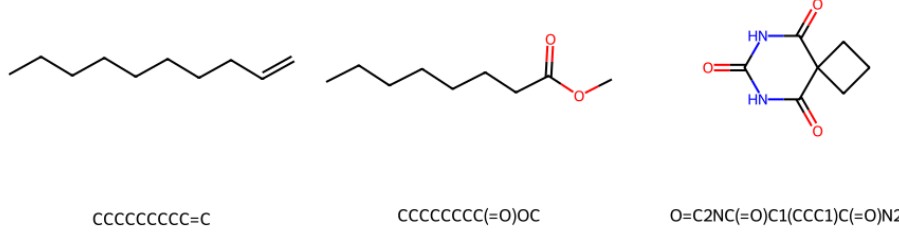

CCCCCCCCC=C            CCCCCCCC(=O)OC            O=C2NC(=O)C1(CCC1)C(=O)N2

Figure 2: Example molecules with their SMILES strings Weininger (1988).

In Figure 3, we show the decomposition of the graph GOSPA similarity. Figure 3a shows the similarity matrix between the molecular graphs of the molecules in Figure 2. The indices 0, 1 and 2 in Figure 3 represent the molecules from left to right in Figure 2.

As can be seen in Figure 2, intuitively, the molecules become more different from left to right, being molecules 0 and 1 more similar than molecule 2. Therefore, the similarity decreases from molecule 0 to molecule 2. Figures 3b, Figure 3c, and Figure 3d show the decomposition of total similarity. In Figure 3b, the matrix shows the similarity in the node elements. By looking at the first row, we can see that molecule 0 is more similar in node elements to molecule 1 than to molecule 2. Figure 3c shows the similarity of the unassigned nodes. Again, molecule 0 is more similar to molecule 1 than to molecule 2 since they have a higher number of assigned nodes. Finally, Figure 3d shows the

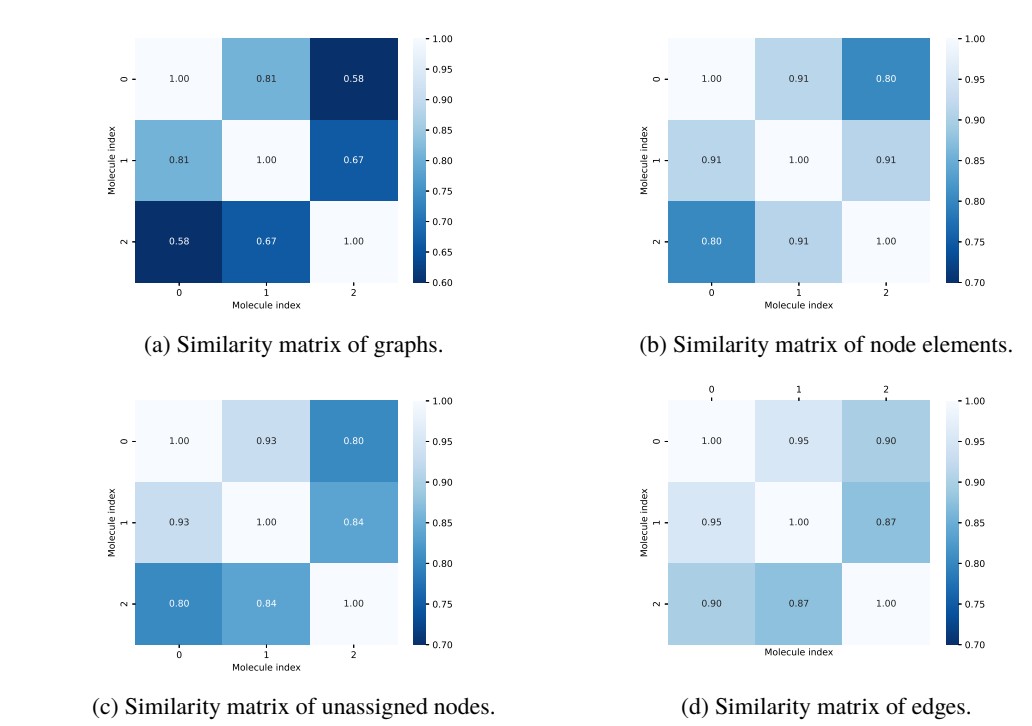

(a) Similarity matrix of graphs.

(b) Similarity matrix of node elements.

(c) Similarity matrix of unassigned nodes.

(d) Similarity matrix of edges.

Figure 3: Plots of the decomposition of the similarity across the three molecules in Figure 2 based on the graph GOSPA similarity. (a) Similarity of graphs; (b) similarity of node elements; (c) similarity of unassigned nodes; (d) similarity of edges.

decomposition for edge similarity between graphs. Again, molecule 0 is more similar to molecule 1 than to molecule 2, since they have a fewer number of edge mismatches.

## 5 CONCLUSION

In this paper, we have proposed a Graph GOSPA similarity function, which is able to measure graph similarity in an interpretable manner based on the graph GOSPA metric. The interpretability that comes from the similarity decomposition is an important characteristics as it helps identify the similar/different aspects between two graphs. We have also introduced a GP model based on the Graph GOSPA similarity, which is able to learn both node and structural features in graphs by measuring differences in node attributes, number of unassigned nodes, and edge mismatches.

Finally, we have evaluated the proposed Graph GOSPA GP on various molecular property prediction datasets. Experimental results demonstrate that Graph GOSPA GP has better performance than the baselines in a number of datasets, and closely follows the best performing algorithms when it does not provide the best results. It has also been the best method at quantifying uncertainty via the NLPD.

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
