# OpenReview forum: "Graph GOSPA Similarity Function for Gaussian Process Regression on Graphs"
_ICLR.cc/2025/Conference — ICLR 2025 Conference Withdrawn Submission_

### Official Review · Reviewer_XZPc · 2024-11-01

**Soundness:** 2
**Presentation:** 2
**Contribution:** 1
**Rating:** 3
**Confidence:** 3

**Summary:**

This paper introduces a graph similarity function based on the graph generalized optimal subpattern assignment (GOSPA) metric, which compares graphs of different sizes by matching nodes optimally. The method was also used in Gaussian processes (GPs) for graphs to predict molecular properties. Experimental results highlight the benefits of the proposed GP model.

**Strengths:**

1. Comparing graphs with unequal sizes
2. Extensive validation on many datasets
3. Exploring connections with Gaussian process

**Weaknesses:**

The paper lacks any theoretical or practical motivation for the proposed metric. The paper reads like "here is the metric, here are some definitions, here is a wide-collection of synthetic datasets, and then our method almost performs well." To be fair, I don't know what to infer from this type of study.

**Questions:**

n/a

---

### Official Review · Reviewer_WinX · 2024-11-02

**Soundness:** 2
**Presentation:** 2
**Contribution:** 1
**Rating:** 3
**Confidence:** 4

**Summary:**

This paper introduces a similarity function for graphs, called the Graph GOSPA similarity, based on the GOSPA metric. By using this similarity function as a kernel in Gaussian Process (GP) regression, the authors create an uncertainty-aware method for graph comparisons. The focus is on molecular property prediction, which the authors demonstrate through experiments on molecular datasets.

**Strengths:**

- **Demonstrated Utility for Molecular Property Prediction**: The method shows competitive performance in molecular property prediction, outperforming a _limited_ number of graph kernel baselines.
- **Uncertainty Quantification**: The algorithm includes inherent uncertainty quantification due to combining GOSPA with Gaussian Processes.
- **Example**: The paper includes a well-visualized example in Section 2.3 and Figure 1, which, though unnecessary lengthy, demonstrates the proposition.
- **Computational Efficiency**: The relaxation of the GOSPA metric enhances computational tractability, making it practical for real-world applications.

**Weaknesses:**

- **Contextualization**: While there is extensive research on assignment problems in statistical graph isomorphisms and graph similarities similarities, the paper lacks a solid contextualization of related work. It does not contrast GOSPA with a sufficiently diverse set of related algorithms. Additionally, contemporary graph similarity algorithms based on Graph Neural Networks (GNNs) are not included.
- **Detailing and Scope**: The paper is light on methodological details, limited in scope, and includes superficial implementation details.
- **Detailing and Scope**: While the descriptions of Gaussian Processes and kernels are sound, they are overly detailed and detract a little from the core focus on developing a GOSPA kernel regression.
- **Omitted Details**: Although key to the paper, the GOSPA metric is described too concisely. Important details have been omitted, which makes it harder to understanding the approach.
- **Minor Contribution**: The paper appears as a minor extension of the GOSPA metric paper [1] with limited additional theoretical or practical value and narrow scope.

  [1] Jinhao Gu, Ángel F. García-Fernández, Robert E. Firth, and Lennart Svensson. Graph GOSPA Metric: A Metric to Measure the Discrepancy Between Graphs of Different Sizes. IEEE Transactions on Signal Processing, 72:4037–4049, 2024.

- **NP-hardness**: The paper claims that computing the GOSPA metric is NP-hard, yet does not offer a rigorous theoretical argument or proof. Instead, it merely states:
  > "Due to the binary constraint in (4), it is NP-hard to compute (5) [*CITATION NEEDED*]."
  This is not generally true for assignment problems. Moreover, as it may be possible to reduce GOSPA to an exact polynomial-time computable assignment problem, avoiding all relaxations.

  Although NP-hardness is claimed, the paper later assumes access to the optimal assignment matrix, which would limit the applicability.
- **Insufficient Theoretical Foundation**: The paper lacks NP-hardness proofs, identifiability results, statistical guarantees, and discussions of the achievable statistical power relative to methods like the Weisfeiler-Leman test.
- **Limited Experimental Scope**: Experiments are limited to small graphs, which are not ideal for testing scalability, and more competitive baselines would be better suited.
- **Applicability Beyond Molecular Graphs**: Although the paper seeks general-purpose applicability for graph similarity, the experiments are limited to a single domain (molecular graphs). Thus, it is unclear whether the method applies efficiently to other types of graphs.
- **Quite Limited Comparisons**: The set of competitors is very limited and focusses on baselines which might not be the ideal choice given the particularities of molecular datasets. Also including recent advances in graph neural networks (GNN) for graph similarities and stronger graph kernels is mandatory for a thorough experimental setup and a solid comparison with the state-of-the-art.
- **Reproducibility**: Code is not accessible during the review process, which limits reproducibility. **However**, since ICLR papers are publicly accessible outside of OpenReview, it is understandable that the authors may want to keep the code private at this stage.

**Questions:**

-/-

---

### Official Review · Reviewer_3q85 · 2024-11-02

**Soundness:** 3
**Presentation:** 3
**Contribution:** 3
**Rating:** 8
**Confidence:** 4

**Summary:**

The paper transforms the Graph GOSPA similarity metric to similarity function as the kernel function of a Gaussian Process. Under a certain condition when one hyper parameter on transformation function and another hyper parameter in similarity metrics are identical, the  decomposability in the similarity function can be achieved. The experiments on multiple datasets for molecule property prediction shows the good prediction accuracy and Uncertainty quantification.

**Strengths:**

1. The Graph GOSPA similarity function has several advantages, including the polynomial computational time, relaxitation, decomposibility.
2. The interpretability is a great feature for molecule property predictions. Through the case study, the paper illustrates clearly how the similarities from different perspectives contribute.
3. Both prediction accuracy and uncertainty are evaluated on multiple real-world datasets to show the effectiveness of the proposed method.

**Weaknesses:**

1. The benefits of the polynomial computational time could be strengthed, especially with Gaussian Process where all the pairwise similarity need to be calculated. The author could include the actual computational time to better illustrate the benefits compared to other methods in the Experiments.

**Questions:**

1. According to Line 221, the choice of hyperparameters are critical to guarantee the positive-semidefinite. How difficult is it to find proper hyperparameters?

---

### Official Review · Reviewer_Sje6 · 2024-11-04

**Soundness:** 2
**Presentation:** 3
**Contribution:** 2
**Rating:** 3
**Confidence:** 3

**Summary:**

This paper suggests converting graph GOSPA distance into a similarity metric and then using it in Gaussian processes to predict molecular properties.

**Strengths:**

The paper is in general well-written and easy to follow. It builds on an interpretable graph distance measure and describes the background and motivation of this measure in detail. The authors report good results compared to other kernels on several molecular datasets.

**Weaknesses:**

The main weakness of the paper is that novelty is limited compared to the existing work by Gu et al. (2024) that introduces the graph GOSPA metric. As I understand, the contributions are: 1) converting an existing distance to a similarity measure (via a standard transformation), 2) using this measure in Gaussian processes, 3) experimental results on molecular datasets.

Many parts of the paper are similar to Gu et al. (2024):
- Section 2.2 describes the graph GOSPA metric, similar to Section III in Gu et al. (2024);
- Figure 1 is similar to Figure 1 in Gu et al. (2024) (part of the caption has the same text);
- The decomposition of the measure in Section 3.3 follows Section IV C in Gu et al. (2024).


Minor comments:
- In many places in the paper \citet is used instead of \citep
- L331: extra space before the footnote

I've also noticed that the style file was modified: the title and section/subsection titles look differently than in the style files.

**Questions:**

It is written in line 218 that "Although trivial similarity functions defined like this are not generally positive semidefinite Vert (2008) ..." I understand that the transformation (10) is not guaranteed to give a valid kernel, but how does this statement follow form Vert (2008)? There, the optimal assignment kernel is analyzed and not the transformation (10).

---

### Note · Authors · 2024-11-13

I have read and agree with the venue's withdrawal policy on behalf of myself and my co-authors.